# Cluster Trees on Manifolds

**Sivaraman Balakrishnan**[†]
sbalakri@cs.cmu.edu

**Srivatsan Narayanan**[†]
srivatsa@cs.cmu.edu

**Alessandro Rinaldo**[‡]
arinaldo@stat.cmu.edu

**Aarti Singh**[†]
aarti@cs.cmu.edu

**Larry Wasserman**[‡]
larry@stat.cmu.edu

**School of Computer Science**[†] **and Department of Statistics**[‡]
Carnegie Mellon University

In this paper we investigate the problem of estimating the cluster tree for a density $f$ supported on or near a smooth $d$-dimensional manifold $M$ isometrically embedded in $\mathbb{R}^D$. We analyze a modified version of a $k$-nearest neighbor based algorithm recently proposed by Chaudhuri and Dasgupta (2010). The main results of this paper show that under mild assumptions on $f$ and $M$, we obtain rates of convergence that depend on $d$ only but not on the ambient dimension $D$. Finally, we sketch a construction of a sample complexity lower bound instance for a natural class of *manifold oblivious* clustering algorithms.

## 1 Introduction

In this paper, we study the problem of estimating the cluster tree of a density when the density is supported on or near a manifold. Let $\mathbf{X} := \{X_1, \ldots, X_n\}$ be a sample drawn i.i.d. from a distribution $P$ with density $f$. The connected components $\mathbb{C}_f(\lambda)$ of the upper level set $\{x : f(x) \geq \lambda\}$ are called *density clusters*. The collection $\mathcal{C} = \{\mathbb{C}_f(\lambda) : \lambda \geq 0\}$ of all such clusters is called the *cluster tree* and estimating this cluster tree is referred to as *density clustering*.

The density clustering paradigm is attractive for various reasons. One of the main difficulties of clustering is that often the true goals of clustering are not clear and this makes clusters, and clustering as a task seem poorly defined. Density clustering however is estimating a well defined population quantity, making its goal, consistent recovery of the *population* density clusters, clear. Typically only mild assumptions are made on the density $f$ and this allows extremely general shapes and numbers of clusters at each level. Finally, the *cluster tree* is an inherently hierarchical object and thus density clustering algorithms typically do not require specification of the "right" level, rather they capture a summary of the density across all levels.

The search for a simple, statistically consistent estimator of the cluster tree has a long history. Hartigan (1981) showed that the popular single-linkage algorithm is not consistent for a sample from $\mathbb{R}^D$, with $D > 1$. Recently, Chaudhuri and Dasgupta (2010) analyzed an algorithm which is both simple and consistent. The algorithm finds the connected components of a sequence of carefully constructed neighborhood graphs. They showed that, as long as the parameters of the algorithm are chosen appropriately, the resulting collection of connected components correctly estimates the cluster tree with high probability.

In this paper, we are concerned with the problem of estimating the cluster tree when the density $f$ is supported on or near a low dimensional manifold. The motivation for this work stems from the problem of devising and analyzing clustering algorithms with provable performance that can be used in high dimensional applications. When data live in high dimensions, clustering (as well as other statistical tasks) generally become prohibitively difficult due to the curse of dimensionality,

which demands a very large sample size. In many high dimensional applications however data is not spread uniformly but rather concentrates around a low dimensional set. This so-called manifold hypothesis motivates the study of data generated on or near low dimensional manifolds and the study of procedures that can adapt effectively to the intrinsic dimensionality of this data.

Here is a brief summary of the main contributions of our paper: (1) We show that the simple algorithm studied in the paper Chaudhuri and Dasgupta (2010) is consistent and has fast rates of convergence for data on or near a low dimensional manifold $M$. The algorithm does not require the user to first estimate $M$ (which is a difficult problem). In other words, the algorithm adapts to the (unknown) manifold. (2) We show that the sample complexity for identifying salient clusters is independent of the ambient dimension. (3) We sketch a construction of a sample complexity lower bound instance for a natural class of clustering algorithms that we study in this paper. (4) We introduce a framework for studying consistency of clustering when the distribution is not supported on a manifold but rather, is concentrated near a manifold. The generative model in this case is that the data are first sampled from a distribution on a manifold and then noise is added. The original data are latent (unobserved). We show that for certain noise models we can still efficiently recover the cluster tree on the *latent* samples.

## 1.1 Related Work

The idea of using probability density functions for clustering dates back to Wishart Wishart (1969). Hartigan (1981) expanded on this idea and formalized the notions of high-density clustering, of the cluster tree and of consistency and fractional consistency of clustering algorithms. In particular, Hartigan (1981) showed that single linkage clustering is consistent when $D = 1$ but is only fractionally consistent when $D > 1$. Stuetzle and R. (2010) and Stuetzle (2003) have also proposed procedures for recovering the cluster tree. None of these procedures however, come with the theoretical guarantees given by Chaudhuri and Dasgupta (2010), which demonstrated that a generalization of Wishart's algorithm allows one to estimate parts of the cluster tree for distributions with full-dimensional support near-optimally under rather mild assumptions. This paper forms the starting point for our work and is reviewed in more detail in the next section.

In the last two decades, much of the research effort involving the use of nonparametric density estimators for clustering has focused on the more specialized problems of optimal estimation of the support of the distribution or of a fixed level set. However, consistency of estimators of a fixed level set does not imply cluster tree consistency, and extending the techniques and analyses mentioned above to hold simultaneously over a variety of density levels is non-trivial. See for example the papers Polonik (1995); Tsybakov (1997); Walther (1997); Cuevas and Fraiman (1997); Cuevas et al. (2006); Rigollet and Vert (2009); Maier et al. (2009); Singh et al. (2009); Rinaldo and Wasserman (2010); Rinaldo et al. (2012), and references therein. Estimating the cluster tree has more recently been considered by Kpotufe and von Luxburg (2011) who also give a simple pruning procedure for removing spurious clusters. Steinwart (2011) and Sriperumbudur and Steinwart (2012) propose procedures for determining recursively the lowest split in the cluster tree and give conditions for asymptotic consistency with minimal assumptions on the density.

## 2 Background and Assumptions

Let $P$ be a distribution supported on an unknown $d$-dimensional manifold $M$. We assume that the manifold $M$ is a $d$-dimensional Riemannian manifold without boundary embedded in a compact set $\mathcal{X} \subset \mathbb{R}^D$ with $d < D$. We further assume that the volume of the manifold is bounded from above by a constant, i.e., $\text{vol}_d(M) \le C$. The main regularity condition we impose on $M$ is that its condition number be not too large. The *condition number* of $M$ is $1/\tau$, where $\tau$ is the largest number such that the open normal bundle about $M$ of radius $r$ is imbedded in $\mathbb{R}^D$ for every $r < \tau$. The condition number is discussed in more detail in the paper Niyogi et al. (2008).

The Euclidean norm is denoted by $\| \cdot \|$ and $v_d$ denotes the volume of the $d$-dimensional unit ball in $\mathbb{R}^d$. $B(x, r)$ denotes the full-dimensional ball of radius $r$ centered at $x$ and $B_M(x, r) := B(x, r) \cap$

$M$. For $Z \subset \mathbb{R}^d$ and $\sigma > 0$, define $Z_\sigma = Z + B(0, \sigma)$ and $Z_{M,\sigma} = (Z + B(0, \sigma)) \cap M$. Note that $Z_\sigma$ is full dimensional, while if $Z \subseteq M$ then $Z_{M,\sigma}$ is $d$-dimensional.

Let $f$ be the density of $P$ with respect to the uniform measure on $M$. For $\lambda \geq 0$, let $\mathbb{C}_f(\lambda)$ be the collection of connected components of the level set $\{x \in \mathcal{X} : f(x) \geq \lambda\}$ and define the *cluster tree* of $f$ to be the hierarchy $\mathcal{C} = \{\mathbb{C}_f(\lambda) : \lambda \geq 0\}$. For a fixed $\lambda$, any member of $\mathbb{C}_f(\lambda)$ is a cluster. For a cluster $C$ its restriction to the sample $\mathbf{X}$ is defined to be $C[\mathbf{X}] = C \cap \mathbf{X}$. The restriction of the cluster tree $\mathcal{C}$ to $\mathbf{X}$ is defined to be $\mathcal{C}[\mathbf{X}] = \{C \cap \mathbf{X} : C \in \mathcal{C}\}$. Informally, this restriction is a dendrogram-like hierarchical partition of $\mathbf{X}$.

To give finite sample results, following Chaudhuri and Dasgupta (2010), we define the notion of salient clusters. Our definitions are slight modifications of those in Chaudhuri and Dasgupta (2010) to take into account the manifold assumption.

**Definition 1** *Clusters $A$ and $A'$ are $(\sigma, \epsilon)$ separated if there exists a nonempty $S \subset M$ such that:*

1. *Any path along $M$ from $A$ to $A'$ intersects $S$.*
2. $\sup_{x \in S_{M,\sigma}} f(x) < (1 - \epsilon) \inf_{x \in A_{M,\sigma} \cup A'_{M,\sigma}} f(x).$

Chaudhuri and Dasgupta (2010) analyze a robust single linkage (RSL) algorithm (in Figure 1). An RSL algorithm estimates the connected components at a level $\lambda$ in two stages. In the first stage, the sample is *cleaned* by thresholding the $k$-nearest neighbor distance of the sample points at a radius $r$ and then, in the second stage, the cleaned sample is *connected* at a connection radius $R$. The connected components of the resulting graph give an estimate of the restriction $\mathbb{C}_f(\lambda)[\mathbf{X}]$. In Section 4 we prove a sample complexity lower bound for the *class of RSL algorithms* which we now define.

**Definition 2** *The* class of RSL algorithms *refers to any algorithm that is of the form described in the algorithm in Figure 1 and relying on Euclidean balls, with any choice of $k$, $r$ and $R$.*

We define two notions of consistency for an estimator $\widehat{\mathcal{C}}$ of the cluster tree:

**Definition 3 (Hartigan consistency)** *For any sets $A$, $A' \subset \mathcal{X}$, let $A_n$ (resp., $A'_n$) denote the smallest cluster of $\widehat{\mathcal{C}}$ containing $A \cap \mathbf{X}$ (resp, $A' \cap \mathbf{X}$). We say $\widehat{\mathcal{C}}$ is consistent if, whenever $A$ and $A'$ are different connected components of $\{x : f(x) \geq \lambda\}$ (for some $\lambda > 0$), the probability that $A_n$ is disconnected from $A'_n$ approaches $1$ as $n \to \infty$.*

**Definition 4 ($(\sigma, \epsilon)$ consistency)** *For any sets $A$, $A' \subset \mathcal{X}$ such that $A$ and $A'$ are $(\sigma, \epsilon)$ separated, let $A_n$ (resp., $A'_n$) denote the smallest cluster of $\widehat{\mathcal{C}}$ containing $A \cap \mathbf{X}$ (resp, $A' \cap \mathbf{X}$). We say $\widehat{\mathcal{C}}$ is consistent if, whenever $A$ and $A'$ are different connected components of $\{x : f(x) \geq \lambda\}$ (for some $\lambda > 0$), the probability that $A_n$ is disconnected from $A'_n$ approaches $1$ as $n \to \infty$.*

The notion of $(\sigma, \epsilon)$ *consistency* is similar that of Hartigan consistency except restricted to $(\sigma, \epsilon)$ separated clusters $A$ and $A'$.

Chaudhuri and Dasgupta (2010) prove a theorem, establishing finite sample bounds for a particular RSL algorithm. In their result there is no manifold and $f$ is a density with respect to the Lebesgue measure on $\mathbb{R}^D$. Their result in essence says that if

$$n \geq O\left(\frac{D}{\lambda\epsilon^2 v_D(\sigma/2)^D} \log \frac{D}{\lambda\epsilon^2 v_D(\sigma/2)^D}\right)$$

then an RSL algorithm with appropriately chosen parameters can resolve any pair of $(\sigma, \epsilon)$ clusters at level at least $\lambda$. It is important to note that this theorem does not apply to the setting when distributions are supported on a lower dimensional set for at least two reasons: (1) the density $f$ is singular with respect to the Lebesgue measure on $\mathcal{X}$ and so the cluster tree is trivial, and (2) the definitions of saliency with respect to $\mathcal{X}$ are typically not satisfied when $f$ has a lower dimensional support.

1. For each $X_i$, $r_k(X_i) := \inf\{r : B(X_i, r) \text{ contains } k \text{ data points}\}$.
2. As $r$ grows from 0 to $\infty$:
    (a) Construct a graph $G_{r,R}$ with nodes $\{X_i : r_k(X_i) \leq r\}$ and edges $(X_i, X_j)$ if $\|X_i - X_j\| \leq R$.
    (b) Let $\mathbb{C}(r)$ be the connected components of $G_{r,R}$.
3. Denote $\widehat{\mathcal{C}} = \{\mathbb{C}(r) : r \in [0, \infty)\}$ and return $\widehat{\mathcal{C}}$.

Figure 1: Robust Single Linkage (RSL) Algorithm

## 3  Clustering on Manifolds

In this section we show that the RSL algorithm can be adapted to recover the cluster tree of a distribution supported on a manifold of dimension $d < D$ with the rates depending only on $d$. In place of the cluster salience parameter $\sigma$, our rates involve a new parameter $\rho$

$$\rho := \min\left(\frac{3\sigma}{16}, \frac{\epsilon\tau}{72d}, \frac{\tau}{16}\right).$$

The precise reason for this definition of $\rho$ will be clear from the proofs (particularly of Lemma 7) but for now notice that in addition to $\sigma$ it is dependent on the condition number $1/\tau$ and deteriorates as the condition number increases. Finally, to succinctly present our results we use $\mu := \log n + d\log(1/\rho)$.

**Theorem 5** *There are universal constants $C_1$ and $C_2$ such that the following holds. For any $\delta > 0$, $0 < \epsilon < 1/2$, run the algorithm in Figure 1 on a sample $\mathbf{X}$ drawn from $f$, where the parameters are set according to the equations*

$$R = 4\rho \quad \text{and} \quad k = C_1 \log^2(1/\delta)(\mu/\epsilon^2).$$

*Then with probability at least $1 - \delta$, $\widehat{\mathcal{C}}$ is $(\sigma, \epsilon)$ consistent. In particular, the clusters containing $A[\mathbf{X}]$ and $A'[\mathbf{X}]$, where $A$ and $A'$ are $(\sigma, \epsilon)$ separated, are internally connected and mutually disconnected in $\mathbb{C}(r)$ for $r$ defined by*

$$v_d r^d \lambda = \frac{1}{1 - \epsilon/6}\left(\frac{k}{n} + \frac{C_2 \log(1/\delta)}{n}\sqrt{k\mu}\right)$$

*provided $\lambda \geq \frac{2}{v_d \rho^d}\frac{k}{n}$.*

Before we prove this theorem a few remarks are in order:

1. To obtain an explicit sample complexity we plug in the value of $k$ and solve for $n$ from the inequality restricting $\lambda$. The sample complexity of the RSL algorithm for recovering $(\sigma, \epsilon)$ clusters at level at least $\lambda$ on a manifold $M$ with condition number at most $1/\tau$ is

$$n = O\left(\frac{d}{\lambda\epsilon^2 v_d\rho^d}\log\frac{d}{\lambda\epsilon^2 v_d\rho^d}\right)$$

where $\rho = C\min(\sigma, \epsilon\tau/d, \tau)$. Ignoring constants that depend on $d$ the main difference between this result and the result of Chaudhuri and Dasgupta (2010) is that our results only depend on the manifold dimension $d$ and not the ambient dimension $D$ (typically $D \gg d$). There is also a dependence of our result on $1/(\epsilon\tau)^d$, for $\epsilon\tau \ll \sigma$. In Section 4 we sketch the construction of an instance that suggests that this dependence is not an artifact of our analysis and that the sample complexity of the class of RSL algorithms is at least $n \geq 1/(\epsilon\tau)^{\Omega(d)}$.

2. Another aspect is that our choice of the connection radius $R$ depends on the (typically) unknown $\rho$, while for comparison, the connection radius in Chaudhuri and Dasgupta (2010) is chosen to be

$\sqrt{2}r$. Under the mild assumption that $\lambda \le n^{O(1)}$ (which is satisfied for instance, if the density on $M$ is bounded from above), we show in Appendix A.8 that an identical theorem holds for $R = 4r$. $k$ is the only real tuning parameter of this algorithm whose choice depends on $\epsilon$ and an unknown leading constant.

3. It is easy to see that this theorem also establishes consistency for recovering the entire cluster tree by selecting an appropriate schedule on $\sigma_n, \epsilon_n$ and $k_n$ that ensures that *all* clusters are distinguished for $n$ large enough (see Chaudhuri and Dasgupta (2010) for a formal proof).

Our proofs structurally mirror those in Chaudhuri and Dasgupta (2010). We begin with a few technical results in 3.1. In Section 3.2 we establish $(\sigma, \epsilon)$ consistency by showing that the clusters are mutually disjoint and internally connected. The main technical challenge is that the curvature of the manifold, modulated by its condition number $1/\tau$, limits our ability to resolve the density level sets from a finite sample, by limiting the maximum cleaning and connection radii the algorithm can use. In what follows, we carefully analyze this effect and show that somewhat surprisingly, despite this curvature, essentially the same algorithm is able to adapt to the unknown manifold and produce a consistent estimate of the entire cluster tree. Similar manifold adaptivity results have been shown in classification Dasgupta and Freund (2008) and in non-parametric regression Kpotufe and Dasgupta (2012); Bickel and Li (2006).

## 3.1 Technical results

In our proof, we use the uniform convergence of the empirical mass of Euclidean balls to their true mass. In the full dimensional setting of Chaudhuri and Dasgupta (2010), this follows from standard VC inequalities. To the best of our knowledge however sharp (ambient dimension independent) inequalities for manifolds are unknown. We get around this obstacle by using the insight that, in order to analyze the RSL algorithms, uniform convergence for Euclidean balls around the *sample points* and around a *fixed minimum $s$-net $\mathcal{N}$ of $M$* (for an appropriately chosen $s$) suffice to analyze the RSL algorithm.

Recall, an $s$-net $\mathcal{N} \subseteq M$ is such that every point of $M$ is at a distance at most $s$ from some point in $\mathcal{N}$. Let $\mathcal{B}_{n,\mathcal{N}} := \left\{ B(z,s) \; : \; z \in \mathcal{N} \cup \mathbf{X}, s \ge 0 \right\}$ be the collection of balls whose centers are sample or net points. We now state our uniform convergence lemma. The proof is in Appendix A.3.

**Lemma 6 (Uniform Convergence)** *Assume $k \ge \mu$. Then there exists a constant $C_0$ such that the following holds. For every $\delta > 0$, with probability $> 1 - \delta$, for all $B \in \mathcal{B}_{n,\mathcal{N}}$, we have:*

$$P(B) \ge \frac{C_\delta \mu}{n} \quad \Longrightarrow \quad P_n(B) > 0,$$

$$P(B) \ge \frac{k}{n} + \frac{C_\delta}{n}\sqrt{k\mu} \quad \Longrightarrow \quad P_n(B) \ge \frac{k}{n},$$

$$P(B) \le \frac{k}{n} - \frac{C_\delta}{n}\sqrt{k\mu} \quad \Longrightarrow \quad P_n(B) < \frac{k}{n},$$

*where $C_\delta := 2C_0 \log(2/\delta)$, and $\mu := 1 + \log n + \log|\mathcal{N}| = Cd + \log n + d\log(1/s)$. Here $P_n(B) = |\mathbf{X} \cap B|/n$ denotes the empirical probability measure of $B$, and $C$ is a universal constant.*

Next we provide a tight estimate of the volume of a small ball intersected with $M$. This bounds the distortion of the apparent density due to the curvature of the manifold and is central to many of our arguments. Intuitively, the claim states that the volume is approximately that of a $d$-dimensional Euclidean ball, provided that its radius is small enough compared to $\tau$. The lower bound is based on Lemma 5.3 of Niyogi et al. (2008) while the upper bound is based on a modification of the main result of Chazal (2013).

**Lemma 7 (Ball volumes)** *Assume $r < \tau/2$. Define $S := B(x,r) \cap M$ for a point $x \in M$. Then*

$$\left(1 - \frac{r^2}{4\tau^2}\right)^{d/2} v_d r^d \le \mathrm{vol}_d(S) \le v_d \left(\frac{\tau}{\tau - 2r_1}\right)^d r_1^d,$$

*where $r_1 = \tau - \tau\sqrt{1 - 2r/\tau}$. In particular, if $r \leq \epsilon\tau/72d$ for $0 \leq \epsilon < 1$, then*

$$v_d r^d (1 - \epsilon/6) \leq \mathrm{vol}_d(S) \leq v_d r^d (1 + \epsilon/6).$$

## 3.2 Separation and Connectedness

**Lemma 8 (Separation)** *Assume that we pick $k$, $r$ and $R$ to satisfy the conditions:*

$$r \leq \rho, \qquad\qquad R = 4\rho,$$

$$v_d r^d (1 - \epsilon/6)\lambda \geq \frac{k}{n} + \frac{C_\delta}{n}\sqrt{k\mu}, \qquad v_d r^d (1 + \epsilon/6)\lambda(1 - \epsilon) \leq \frac{k}{n} - \frac{C_\delta}{n}\sqrt{k\mu}.$$

*Then with probability $1 - \delta$, we have: (1) All points in $A_{\sigma-r}$ and $A'_{\sigma-r}$ are kept, and all points in $S_{\sigma-r}$ are removed. (2) The two point sets $A \cap \mathbf{X}$ and $A' \cap \mathbf{X}$ are disconnected in $G_{r,R}$.*

**Proof.** The proof is analogous to the separation proof of Chaudhuri and Dasgupta (2010) with several modifications. Most importantly, we need to ensure that despite the curvature of the manifold we can still resolve the density well enough to guarantee that we can identify and eliminate points in the region of separation.

Throughout the proof, we will assume that the good event in Lemma 6 (uniform convergence for $\mathcal{B}_{n,\mathcal{N}}$) occurs. Since $r \leq \epsilon\tau/72d$, by Lemma 7 $\mathrm{vol}(B_M(x,r))$ is between $v_d r^d (1 - \epsilon/6)$ and $v_d r^d (1 + \epsilon/6)$, for any $x \in M$. So if $X_i \in A \cup A'$, then $B_M(X_i, r)$ has mass at least $v_d r^d (1 - \epsilon/6)\cdot\lambda$. Since this is $\geq \frac{k}{n} + \frac{C_\delta}{n}\sqrt{k\mu}$ by assumption, this ball contains at least $k$ sample points, and hence $X_i$ is kept. On the other hand, if $X_i \in S_{\sigma-r}$, then the set $B_M(X_i, r)$ contains mass at most $v_d r^d (1 + \epsilon/6) \cdot \lambda(1 - \epsilon)$. This is $\leq \frac{k}{n} - \frac{C_\delta}{n}\sqrt{k\mu}$. Thus by Lemma 6 $B_M(X_i, r)$ contains fewer than $k$ sample points, and hence $X_i$ is removed.

To prove the graph is disconnected, we first need a bound on the geodesic distance between two points that are at most $R$ apart in Euclidean distance. Such an estimate follows from Proposition 6.3 in Niyogi et al. (2008) who show that if $\|p - q\| = R \leq \tau/2$, then the geodesic distance $d_M(p, q) \leq \tau - \tau\sqrt{1 - \frac{2R}{\tau}}$. In particular, if $R \leq \tau/4$, then $d_M(p, q) < R\left(1 + \frac{4R}{\tau}\right) \leq 2R$. Now, notice that if the graph is connected there must be an edge that connects two points that are at a geodesic distance of at least $2(\sigma - r)$. Any path between a point in $A$ and a point in $A'$ along $M$ must pass through $S_{\sigma-r}$ and must have a geodesic length of at least $2(\sigma - r)$. This is impossible if the connection radius satisfies $2R < 2(\sigma - r)$, which follows by the assumptions on $r$ and $R$. $\square$

All the conditions in Lemma 8 can be simultaneously satisfied by setting $k := 16C_\delta^2(\mu/\epsilon^2)$, and

$$v_d r^d (1 - \epsilon/6) \cdot \lambda = \frac{k}{n} + \frac{C_\delta}{n}\sqrt{k\mu}. \tag{1}$$

The condition on $r$ is satisfied since $\lambda \geq \frac{2}{v_d\rho^d}\frac{k}{n}$ and the condition on $R$ is satisfied by its definition.

**Lemma 9 (Connectedness)** *Assume that the parameters $k, r$ and $R$ satisfy the separation conditions (in Lemma 8). Then, with probability at least $1 - \delta$, $A[\mathbf{X}]$ is connected in $G_{r,R}$.*

**Proof.** Let us show that any two points in $A \cap \mathbf{X}$ are connected in $G_{r,R}$. Consider $y, y' \in A \cap \mathbf{X}$. Since $A$ is connected, there is a path $P$ between $y, y'$ lying entirely inside $A$, i.e., a continuous map $P : [0, 1] \to A$ such that $P(0) = y$ and $P(1) = y'$. We can find a sequence of points $y_0, \ldots, y_t \in P$ such that $y_0 = y$, $y_t = y'$, and the geodesic distance on $M$ (and hence the Euclidean distance) between $y_{i-1}$ and $y_i$ is at most $\eta$, for an arbitrarily small constant $\eta$.

Let $\mathcal{N}$ be minimal $R/4$-net of $M$. There exist $z_i \in \mathcal{N}$ such that $\|y_i - z_i\| \leq R/4$. Since $y_i \in A$, we have $z_i \in A_{M,R/4}$, and hence the ball $B_M(z_i, R/4)$ lies completely inside $A_{M,R/2} \subseteq A_{M,\sigma-r}$. In particular, the density inside the ball is at least $\lambda$ everywhere, and hence the mass inside it is at least

$$v_d(R/4)^d(1 - \epsilon/6)\lambda \geq \frac{C_\delta\mu}{n}.$$

Observe that $R \geq 4r$ and so this condition is satisfied as a consequence of satisfying Equation 1. Thus Lemma 6 guarantees that the ball $B_M(z_i, R/4)$ contains at least one sample point, say $x_i$. (Without loss of generality, we may assume $x_0 = y$ and $x_t = y'$.) Since the ball lies completely in $A_{M,\sigma-r}$, the sample point $x_i$ is not removed in the cleaning step (Lemma 8).

Finally, we bound $d(x_{i-1}, x_i)$ by considering the sequence of points $(x_{i-1}, z_{i-1}, y_{i-1}, y_i, z_i, x_i)$. The pair $(y_{i-1}, y_i)$ are at most $s$ apart and the other successive pairs at most $R/4$ apart, hence $d(x_{i-1}, x_i) \leq 4(R/4) + \eta = R + \eta$. The claim follows by letting $\eta \to 0$. $\square$

# 4   A lower bound instance for the class of RSL algorithms

Recall that the sample complexity in Theorem 5 scales as $n = O\left(\frac{d}{\lambda \epsilon^2 v_d \rho^d} \log \frac{d}{\lambda \epsilon^2 v_d \rho^d}\right)$ where $\rho = C \min\left(\sigma, \epsilon\tau/d, \tau\right)$. For full dimensional densities, Chaudhuri and Dasgupta (2010) showed the information theoretic lower bound $n = \Omega\left(\frac{1}{\lambda \epsilon^2 v_D \sigma^D} \log \frac{1}{\lambda \epsilon^2 v_D \sigma^D}\right)$. Their construction can be straightforwardly modified to a $d$-dimensional instance on a smooth manifold. Ignoring constants that depend on $d$, these upper and lower bounds can still differ by a factor of $1/(\epsilon\tau)^d$, for $\epsilon\tau \ll \sigma$. In this section we provide an informal sketch of a hard instance for the class of RSL algorithms (see Definition 2) that suggests a sample complexity lower bound of $n \geq 1/(\epsilon\tau)^{\Omega(d)}$.

We first describe our lower bound instance. The manifold $M$ consists of two disjoint components, $C$ and $C'$ (whose sole function is to ensure $f$ integrates to 1). The component $C$ in turn contains three parts, which we call 'top', 'middle', and 'bottom' respectively. The middle part, denoted $M_2$, is the portion of the standard $d$-dimensional unit sphere $\mathbb{S}^d(0,1)$ between the planes $x_1 = +\sqrt{1-4\tau^2}$ and $x_1 = -\sqrt{1-4\tau^2}$. The top part, denoted $M_1$, is the upper hemisphere of radius $2\tau$ centered at $(+\sqrt{1-4\tau^2}, 0, 0, \ldots, 0)$. The bottom part, denoted $M_3$, is a symmetric hemisphere centered at $(-\sqrt{1-4\tau^2}, 0, 0, \ldots, 0)$. Thus $C$ is obtained by gluing a portion of the unit sphere with two (small) hemispherical caps. $C$ as described does not have a condition number at most $1/\tau$ because of the "corners" at the intersection of $M_2$ and $M_1 \cup M_3$. This can be fixed without affecting the essence of the construction by smoothing this intersection by rolling a ball of radius $\tau$ around it (a similar construction is made rigorous in Theorem 6 of Genovese et al. (2012)). Let $P$ be the distribution on $M$ whose density over $C$ is $\lambda$ if $|x_1| > 1/2$, and $\lambda(1-\epsilon)$ if $|x_1| \leq 1/2$, where $\lambda$ is chosen small enough such that $\lambda \operatorname{vol}_d(C) \leq 1$. The density over $C'$ is chosen such that the total mass of the manifold is 1. Now $M_1$ and $M_3$ are $(\sigma, \epsilon)$ separated at level $\lambda$ for $\sigma = \Omega(1)$. The separator set $S$ is the equator of $M_2$ in the plane $x_1 = 0$.

We now provide some intuition for why RSL algorithms will require $n \geq 1/(\epsilon\tau)^{\Omega(d)}$ to succeed on this instance. We focus our discussion on RSL algorithms with $k > 2$, i.e. on algorithms that do in fact use a *cleaning* step, ignoring the single linkage algorithm which is known to be inconsistent for full dimensional densities. Intuitively, because of the curvature of the described instance, the mass of a sufficiently large Euclidean ball in the separator set is *larger* than the mass of a corresponding ball in the true clusters. This means that any algorithm that uses large balls cannot reliably clean the sample and this restricts the size of the balls that can be used. Now if points in the regions of high density are to survive then there must be $k$ sample points in the *small* ball around any point in the true clusters and this gives us a lower bound on the necessary sample size.

The RSL algorithms work by counting the number of sample points inside the balls $B(x, r)$ centered at the sample points $x$, for some radius $r$. In order for the algorithm to reliably resolve $(\sigma, \epsilon)$ clusters, it should distinguish points in the separator set $S \subset M_2$ from those in the level $\lambda$ clusters $M_1 \cup M_3$. A necessary condition for this is that the mass of a ball $B(x, r)$ for $x \in S_{\sigma-r}$ should be strictly smaller than the mass inside $B(y, r)$ for $y \in M_1 \cup M_3$. In Appendix A.4, we show that this condition restricts the radius $r$ to be at most $O(\tau\sqrt{\epsilon/d})$. Now, consider any sample point $x_0$ in $M_1 \cup M_3$ (such an $x$ exists with high probability). Since $x_0$ should not be removed during the cleaning step, the ball $B(x_0, r)$ must contain some other sample point (indeed, it must contain at least $k-1$ more sample points). By a union bound, this happens with probability at most $(n-1)v_d r^d \lambda \leq$

$O(d^{-d/2}n\tau^d\epsilon^{d/2}\lambda)$. If we want the algorithm to succeed with probability at least $1/2$ (say) then $n \geq \Omega\left(\frac{d^{d/2}}{\tau^d\lambda\epsilon^{d/2}}\right)$.

## 5 Cluster tree recovery in the presence of noise

So far we have considered the problem of recovering the cluster tree given samples from a density supported *on* a lower dimensional manifold. In this section we extend these results to the more general situation when we have *noisy* samples concentrated *near* a lower dimensional manifold. Indeed it can be argued that the manifold + noise model is a natural and general model for high-dimensional data. In the noisy setting, it is clear that we can infer the cluster tree of the *noisy* density in a straightforward way. A stronger requirement would be consistency with respect to the underlying *latent* sample. Following the literature on manifold estimation (Balakrishnan et al. (2012); Genovese et al. (2012)) we consider two main noise models. For both of them, we specify a distribution $Q$ for the noisy sample.

**1. Clutter Noise:** We observe data $Y_1, \ldots, Y_n$ from the mixture $Q := (1 - \pi)U + \pi P$ where $0 < \pi \leq 1$ and $U$ is a uniform distribution on $\mathcal{X}$. Denote the samples drawn from $P$ in this mixture $\mathbf{X} = \{X_1, \ldots, X_m\}$. The points drawn from $U$ are called background clutter. In this case, we can show:

**Theorem 10** *There are universal constants $C_1$ and $C_2$ such that the following holds. For any $\delta > 0$, $0 < \epsilon < 1/2$, run the algorithm in Figure 1 on a sample $\{Y_1, \ldots, Y_n\}$, with parameters*

$$R := 4\rho \quad k := C_1 \log^2(1/\delta)(\mu/\epsilon^2).$$

*Then with probability at least $1 - \delta$, $\widehat{\mathcal{C}}$ is $(\sigma, \epsilon)$ consistent. In particular, the clusters containing $A[\mathbf{X}]$ and $A'[\mathbf{X}]$ are internally connected and mutually disconnected in $\mathbb{C}(r)$ for $r$ defined by*

$$\pi v_d r^d \lambda = \frac{1}{1 - \epsilon/6}\left(\frac{k}{n} + \frac{C_2 \log(1/\delta)}{n}\sqrt{k\mu}\right)$$

*provided $\lambda \geq \max\left\{\frac{2}{v_d\rho^d}\frac{k}{n}, \frac{2v_D^{d/D}(1-\pi)^{d/D}}{v_d\epsilon^{d/D}\pi}\left(\frac{k}{n}\right)^{1-d/D}\right\}$ where $\rho$ is now slightly modified (in constants), i.e., $\rho := \min\left(\frac{\sigma}{7}, \frac{\epsilon\tau}{72d}, \frac{\tau}{24}\right)$.*

**2. Additive Noise**: The data are of the form $Y_i = X_i + \eta_i$ where $X_1, \ldots, X_n \sim P$, and $\eta_1, \ldots, \eta_n$ are a sample from *any* bounded noise distribution $\Phi$, with $\eta_i \in B(0, \theta)$. Note that $Q$ is the convolution of $P$ and $\Phi$, $Q = P \star \Phi$.

**Theorem 11** *There are universal constants $C_1$ and $C_2$ such that the following holds. For any $\delta > 0$, $0 < \epsilon < 1/2$, run the algorithm in Figure 1 on the sample $\{Y_1, \ldots, Y_n\}$ with parameters*

$$R := 5\rho \quad k := C_1 \log^2(1/\delta)(\mu/\epsilon^2).$$

*Then with probability at least $1 - \delta$, $\widehat{\mathcal{C}}$ is $(\sigma, \epsilon)$ consistent for $\theta \leq \rho\epsilon/24d$. In particular, the clusters containing $\{Y_i : X_i \in A\}$ and $\{Y_i : X_i \in A'\}$ are internally connected and mutually disconnected in $\mathbb{C}(r)$ for $r$ defined by*

$$v_d r^d(1 - \epsilon/12)(1 - \epsilon/6)\lambda = \frac{k}{n} + \frac{C_\delta}{n}\sqrt{k\mu}$$

*if $\lambda \geq \frac{2}{v_d\rho^d}\frac{k}{n}$ and $\theta \leq \rho\epsilon/24d$, where $\rho := \min\left(\frac{\sigma}{7}, \frac{\tau}{24}, \frac{\epsilon\tau}{144d}\right)$.*

The proofs for both Theorems 10 and 11 appear in Appendix A.5. Notice that in each case we receive samples from a *full* $D$-dimensional distribution but are still able to achieve rates independent of $D$ because these distributions are concentrated around the lower dimensional $M$. For the clutter noise case we produce a tree that is consistent for samples drawn from $P$ (which are *exactly* on $M$), while in the additive noise case we produce a tree on the observed $Y_i$s which is $(\sigma, \epsilon)$ consistent for the *latent* $X_i$s (for $\theta$ small enough). It is worth noting that in the case of clutter noise we can still consistently recover the *entire* cluster tree. Intuitively, this is because the $k$-NN distances for points on $M$ are much smaller than for clutter points that are far away from $M$. As a result the clutter noise only affects a vanishingly low level set of the cluster tree.

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
