[Supplementary Material]

# A    Additional proofs

In this section we first prove some technical lemmas before giving full proofs of various claims made in the paper.

## A.1    Volume estimates for small balls on manifolds

**Theorem 12** *If*

$$r \leq \frac{\epsilon \tau}{12d}$$

*for $0 \leq \epsilon < 1$ then*

$$v_d r^d \left(1 - \epsilon\right) \leq \text{vol}(S) \leq v_d r^d \left(1 + \epsilon\right).$$

**Proof.** The lower bound follows from Niyogi et al. (2008) (Lemma 5.3) who show that

**Lemma 13** *For $r < \frac{\tau}{2}$*

$$\text{vol}(S) \geq \left(1 - \frac{r^2}{4\tau^2}\right)^{d/2} v_d r^d.$$

The upper bound follows from Chazal (2013) who shows that

**Lemma 14** *For $r < \frac{\tau}{2}$*

$$\text{vol}(S) \leq v_d \left(\frac{\tau}{\tau - 2\alpha}\right)^d \alpha^d$$

*where*

$$\alpha = \tau - \tau \sqrt{1 - \frac{2r}{\tau}}.$$

To produce the result of the theorem we will need some careful manipulation of these two lemmas. In particular, we need the following estimates

**Lemma 15**
$$f(x) = (1 - x)^{1/2} \geq 1 - \frac{x}{2} - x^2$$

*if $0 \leq x \leq \frac{1}{2}$.*

$$f(x) = (1 + x)^n \leq 1 + 2nx$$

*if $0 \leq x \leq \frac{1}{2n}$.*

$$f(x) = (1 - x)^{-1} \leq 1 + 2x$$

*if $0 \leq x \leq 1/2$.*

$$f(x) = (1 - x)^n \geq 1 - 2nx$$

*if $0 \leq x \leq \frac{1}{2n}$.*

The proof of this lemma is straightforward based on approximations via Taylor's series and we omit them.

Using Lemma 15 we have

$$\alpha \leq r \left(1 + \frac{4r}{\tau}\right)$$

if $r \leq \frac{\tau}{4}$. Now, using this also notice that

$$\frac{\tau}{\tau - 2\alpha} \leq \frac{1}{1 - \frac{2r}{\tau}\left(1 + \frac{4r}{\tau}\right)} \leq 1 + \frac{4r}{\tau}\left(1 + \frac{4r}{\tau}\right)$$

where the second inequality follows from Lemma 15 if $r \leq \tau/8$.

Combining these we have the following:

for all $r \leq \frac{\tau}{8}$

$$v_d r^d \left(1 - \frac{r^2}{4\tau^2}\right)^{d/2} \leq \text{vol}(S) \leq v_d r^d \left(1 + \frac{6r}{\tau}\right)^d$$

The final result now follows another application of Lemma 15 on each side of this inequality. $\square$

## A.2 Bound on covering number

We need the following bound on the covering number of a manifold. See the paper Niyogi et al. (2008) (p. 16) for a proof.

**Lemma 16** *For $s \leq 2\tau$, the $s$-covering number of $M$ is at most*

$$\frac{\text{vol}_d(M)}{\cos^d(\arcsin(s/4\tau))v_d(s/2)^d} \leq O\left(\frac{\text{vol}_d(M)c^d}{v_d s^d}\right)$$

*for an absolute constant $c$. In particular, if $\text{vol}_d(M)$ is bounded above by a constant, the $s$-covering number of $M$ is at most $O(c^d/(v_d s^d))$.*

**Proof.** We prove only the second claim. For $s \leq 2\tau$, we have $\arcsin(s/4\tau) \leq \pi/6$, and hence $\cos(\arcsin(s/4\tau)) \geq \sqrt{3}/2$. Plugging this in the bound, we get

$$|\mathcal{N}| \leq \frac{\text{vol}_d(M)(2/\sqrt{3})^d}{v_d(s/2)^d},$$

which gives the claim with $c = 4/\sqrt{3}$. $\square$

## A.3 Uniform convergence

In this subsection, we prove uniform convergence for balls centered on sample and net points (Lemma 6). Consider the family of balls centered at a fixed point $z$, $\mathcal{B}_z := \left\{B(z,s) \ : \ s \geq 0\right\}$. This collection has VC dimension 1. Thus with probability $1 - \delta'$, it holds that for every $B \in \mathcal{B}_z$, we have

$$\max\left\{\frac{P(B) - P_n(B)}{\sqrt{P(B)}}, \frac{P(B) - P_n(B)}{\sqrt{P_n(B)}}\right\} \leq 2\sqrt{\frac{\log(2n) + \log(4/\delta')}{n}},$$

where $P(B)$ is the true mass of $B$, and $P_n(B) = |\mathbf{X} \cap B|/n$ is its empirical measure. By a union bound over all $z \in \mathcal{N}$, setting $\delta' := \delta/(2|\mathcal{N}|)$, the following holds uniformly for every $z \in \mathcal{N}$ and every $B \in \mathcal{B}_z$ with probability $1 - \delta/2$:

$$\max\left\{\frac{P(B) - P_n(B)}{\sqrt{P(B)}}, \frac{P(B) - P_n(B)}{\sqrt{P_n(B)}}\right\} \leq 2\sqrt{\frac{\log(2n) + \log(8|\mathcal{N}|/\delta)}{n}}.$$

To provide a similar uniform convergence result for balls centered at a sample point $X_i$, we consider the $(n-1)$-subsample $X_i^{n-1}$ of $\mathbf{X}$ obtained by deleting $X_i$ from the sample. Let $P_i^{n-1}$ be the empirical probability measure of this subsample:

$$P_{n-1}(B) := \frac{1}{n-1}\sum_{j \neq i}\mathbb{I}[X_i \in B].$$

It is easy to check that $P_{n-1}$ is uniformly close to $P_n$. In particular, for every set $B$ containing $X_i$, we have

$$P_{n-1}(B) \leq P_n(B) \leq P_{n-1}(B) + \frac{1}{n}. \tag{2}$$

Now, with probability at least $1 - \delta/(2n)$, for any ball $B$ centered at $X_i$,

$$P(B) - P_{n-1}(B) \leq 2\sqrt{\frac{\log(2n-2) + \log 8n/\delta}{n-1}} \cdot \sqrt{P(B)},$$

$$P_{n-1}(B) - P(B) \leq 2\sqrt{\frac{\log(2n-2) + \log 8n/\delta}{n-1}} \cdot \sqrt{P_{n-1}(B)}.$$

Using (2), we get

$$P(B) - P_n(B) \leq 2\sqrt{\frac{\log(2n-2) + \log 8n/\delta}{n-1}} \cdot \sqrt{P(B)},$$

$$P_n(B) - P(B) \leq 2\sqrt{\frac{\log(2n-2) + \log 8n/\delta}{n-1}} \cdot \sqrt{P_n(B)} + \frac{1}{n}.$$

By a union bound over all $X_i \in \mathbf{X}$, we get the claimed inequalities for all sample points with probability $1 - \delta/2$.

Putting together our bounds for balls around sample and net points, with probability at least $1 - \delta$, it holds that for all $B \in \mathcal{B}_{n,\mathcal{N}}$, we have

$$P(B) - P_n(B) \leq O\left(\sqrt{\frac{\mu + \log(1/\delta)}{n}}\right) \cdot \sqrt{P(B)},$$

$$P_n(B) - P(B) \leq O\left(\sqrt{\frac{\mu + \log(1/\delta)}{n}}\right) \cdot \sqrt{P_n(B)} + \frac{1}{n}.$$

for $\mu = 1 + \log n + \log|\mathcal{N}| = O(d) + \log n + d\log(1/s)$ (using Lemma 16). The lemma now follows using simple manipulations of these inequalities (see Chaudhuri and Dasgupta (2010) for details).

### A.4 Sketch of the lower bound instance

The following lemma gives an estimate of the volume of the intersection of a small ball with a sphere.

**Lemma 17 (Volume of a spherical cap)** *Suppose $\mathbb{S}^d$ is a $d$-dimensional sphere of radius $\tau$ (embedded in $\mathbb{R}^{d+1}$), and let $x \in \mathbb{S}^d$. Then, for small enough $r$, it holds that*

$$\mathrm{vol}_d(B(x,r) \cap \mathbb{S}^d) = v_d r^d \left(1 - c_d \frac{r^2}{\tau^2} + O_d\left(\frac{r^4}{\tau^4}\right)\right)$$

*where $c_d := \frac{d(d-2)}{8(d+2)}$. Note that $c_1 < 0$, $c_2 = 0$, and $c_d > 0$ for all $d \geq 3$.*

In this section, we prove Lemma 17. The height $h$ of the cap can be easily checked to be equal to $h = r^2/2\tau$. Now, the volume of the cap is given by the formula

$$v_{cap} = \frac{\pi^{(d+1)/2}\tau^d}{\Gamma((d+1)/2)} I_\alpha(d/2, 1/2)$$

where the parameter $\alpha$ is defined by

$$\alpha := \frac{2\tau h - h^2}{\tau} = \frac{r^2}{\tau^2}\left(1 - \frac{r^2}{4\tau^2}\right).$$

Further $I_\alpha(\cdot, \cdot)$ represents the incomplete beta function:

$$\begin{aligned}
I_\alpha(z, w) &= \frac{B(\alpha; z, w)}{B(z, w)} \\
&= \frac{\int_0^\alpha u^{z-1}(1-u)^{w-1}du}{B(z, w)} \\
&= \frac{\Gamma(z+w)}{\Gamma(z)\Gamma(w)}\int_0^\alpha u^{z-1}(1-u)^{w-1}du.
\end{aligned}$$

Thus,

$$
\begin{aligned}
v_{cap} &= \frac{\pi^{(d+1)/2}\tau^d}{\Gamma((d+1)/2)} \cdot \frac{\Gamma((d+1)/2))}{\Gamma(d/2)\Gamma(1/2)} \cdot \int_0^\alpha u^{d/2-1}(1-u)^{-1/2}du \\
&= \frac{\pi^{d/2}\tau^d}{\Gamma(d/2)} \int_0^\alpha u^{d/2-1}(1-u)^{-1/2}du \\
&= \frac{dv_d\tau^d}{2} \int_0^\alpha u^{d/2-1}(1-u)^{-1/2}du.
\end{aligned}
$$

Since $\alpha \to 0$ as $r \to 0$, we can approximate the integral by expanding the integrand as a Taylor series around 0:

$$
\begin{aligned}
v_{cap} &= \frac{dv_d\tau^d}{2} \int_0^\alpha u^{d/2-1}\left(1 + u/2 + O(u^2)\right)du \\
&= \frac{dv_d\tau^d}{2} \left(\frac{\alpha^{d/2}}{d/2} + \frac{1}{2}\frac{\alpha^{d/2+1}}{d/2+1} + O(\alpha^{d/2+2})\right) \\
&= v_d\tau^d\alpha^{d/2}\left(1 + \frac{d}{2(d+2)}\alpha + O(\alpha^2))\right).
\end{aligned}
$$

Finally, using $\alpha := \frac{r^2}{\tau^2}(1 - \frac{r^2}{\tau^2})$, we get

$$
\begin{aligned}
v_{cap} &= v_d r^d \left(1 - \frac{r^2}{4\tau^2}\right)^{d/2}\left(1 + \frac{dr^2}{2(d+2)\tau^2} + O\left(\frac{r^4}{\tau^4}\right)\right) \\
&= v_d r^d \cdot \left(1 - \frac{dr^2}{8\tau^2} + \frac{dr^2}{2(d+2)\tau^2} + O_d\left(\frac{r^4}{\tau^4}\right)\right),
\end{aligned}
$$

which simplifies to the claimed estimate.

We now show that it must be the case that $r \le O(\tau\sqrt{\epsilon/d})$. We argued that for the algorithm to reliably resolve the $(\sigma, \epsilon)$ separated clusters $M_1$ and $M_3$, an $r$-ball around a sample point in $S_{\sigma-r}$ must have mass appreciably smaller than those around points in $M_1$. By the previous lemma, the two kinds of balls have volumes

$$
v_d r^d \left(1 - c_d\frac{r^2}{1^2} + O_d\left(\frac{r^4}{1^4}\right)\right) = v_d r^d \left(1 - c_d r^2 + O_d(r^4)\right)
$$

and

$$
v_d r^d \left(1 - c_d\frac{r^2}{4\tau^2} + O_d\left(\frac{r^4}{16\tau^4}\right)\right) = v_d r^d \left(1 - c_d\frac{r^2}{4\tau^2} + O_d\left(\frac{r^4}{\tau^4}\right)\right).
$$

Thus we must have

$$
v_d r^d v_d r^d \left(1 - c_d r^2 + O_d(r^4)\right) \cdot \lambda(1 - \epsilon) \le v_d r^d \left(1 - c_d\frac{r^2}{4\tau^2} + O_d\left(\frac{r^4}{\tau^4}\right)\right) \cdot \lambda.
$$

This implies that $r^2 \le O\left(\frac{4\tau^2\epsilon}{(1-4\tau^2)c_d}\right)$. Hence if $\tau \le 1/4$, we have $r \le \tau\sqrt{\epsilon/c_d}$. Plugging in $c_d = \Omega(d)$ gives us the claim.

## A.5 Clustering with noisy samples

## A.6 Proof of Theorem 10

As before we begin by showing separation followed by a proof of connectivity. Recall that $\rho := \min\left(\frac{\sigma}{7}, \frac{\epsilon\tau}{72d}, \frac{\tau}{24}\right)$.

**Lemma 18 (Separation)** *Assume that we pick $k$, $r$ and $R$ to satisfy the conditions:*

$$r \leq \rho, \quad R = 4\rho$$

$$\pi \cdot v_d r^d (1 - \epsilon/6) \cdot \lambda \geq \frac{k}{n} + \frac{C_\delta}{n} \sqrt{k\mu},$$

$$\pi \cdot v_d r^d (1 + \epsilon/6) \cdot \lambda(1 - \epsilon) + (1 - \pi) \cdot v_D r^D \leq \frac{k}{n} - \frac{C_\delta}{n} \sqrt{k\mu}.$$

*Then with probability $1 - \delta$, it holds that:*

1. *All points in $A_{M,\sigma-r}$ and $A'_{M,\sigma-r}$ are kept, and all points in $\mathcal{X} \backslash M_r$ and $S_{\sigma-r}$ are removed. Here, $M_r$ is the tubular region around $M$ of width $r$.*

2. *The two point sets $A[\mathbf{X}]$ and $A'[\mathbf{X}]$ are disconnected in the graph $G_{r,R}$.*

**Proof.** The proof of the first claim is similar to the noiseless setting, except that the probability mass inside a ball now has contributions from both the manifold and the background clutter. For $x \in S_{\sigma-r}$, the probability mass of the ball $B(x,r)$ under $Q$ is at most $\pi v_d r^d (1 + \epsilon/6) \cdot \lambda(1 - \epsilon) + (1 - \pi)v_D r^D$, which is at most $\frac{k}{n} - \frac{C_\delta}{n}\sqrt{k\mu}$. Thus $x$ is removed during the cleaning step. Similarly, if $x \notin M_r$, the ball $B(x,r)$ does not intersect the manifold, and hence its mass is at most $(1 - \pi)v_D r^D$. Hence all points outside $M_r$ are removed. Finally, if $x \in (A_{M,\sigma-r} \cup A'_{M,\sigma-r}) \cap \mathbf{X}$, then the mass of the ball $B_M(x,r)$ is at least $v_d r^d (1 - \epsilon/6)\lambda$ (ignoring the contribution of the noise). This is at least $\frac{k}{n} + \frac{C_\delta}{n}\sqrt{k\mu}$, and hence $x$ is kept.

To prove the second claim, suppose that sets $A \cap \mathbf{X}$ and $A' \cap \mathbf{X}$ are connected in $G_{r,R}$. Then there exists a sequence of sample points $y_0, y_1, \ldots, y_t$ such that $y_0 \in A$, $y_t \in A'$ and $d(y_{i-1}, y_i) \leq R$ for all $1 \leq i \leq t$. Let $x_i$ be the projection of $y_i$ on $M$, i.e., $x_i$ is the point of $M$ closest to $y_i$. We have already showed that each $y_i$ lies inside the tube $M_r$, so $d(x_i, y_i) \leq r$, and hence by triangle inequality, we have $d(x_{i-1}, x_i) \leq R + 2r \leq \tau/4$. Hence, the geodesic distance between $x_{i-1}$ and $x_i$ is $< 2(R + 2r)$. Now, by an argument analogous to the noiseless setting, there exists a pair $(x_{i-1}, x_i)$ which are at a (geodesic) distance at least $2(\sigma - r)$. This is a contradiction since our parameter setting implies that $2(\sigma - r) \geq 2(R + 2r)$. $\square$

**Lemma 19 (Connectedness)** *Assume that the parameters $k, r$ and $R$ satisfy the separation conditions (in Lemma 18). Then, with probability at least $1 - \delta$, $A \cap \mathbf{Y}$ is connected in $G_{r,R}$.*

**Proof.** The proof of this lemma is identical to Lemma 9 and is omitted. $\square$

We now show how to pick the parameters to satisfy the conditions in Lemma 18. Set $k := 144C_\delta^2(\mu/\epsilon^2)$, and define $r$ by

$$\pi v_d r^d (1 - \epsilon/6) \cdot \lambda = \frac{k}{n} + \frac{C_\delta}{n}\sqrt{k\mu}.$$

It is easy to check that this setting satisfies all our requirements, provided that the term $(1 - \pi)v_D r^D$ arising from the clutter noise satisfies the additional constraint

$$(1 - \pi)v_D r^D \leq (\epsilon/2) \times \pi v_d r^d \lambda.$$

The definition of $r$ implies that $r$ is upper bounded by $\left(\frac{2k}{n\lambda\pi v_d}\right)^{1/d}$. Thus it suffices to ensure that

$$(1 - \pi)v_D \left(\frac{2k}{n\lambda\pi v_d}\right)^{D/d} \leq (\epsilon/2) \cdot \frac{2k}{n} = \frac{k\epsilon}{n}.$$

This is equivalent to the condition

$$\lambda \geq \frac{2v_D^{d/D}}{v_d \epsilon^{d/D}} \cdot \frac{(1 - \pi)^{d/D}}{\pi} \cdot \left(\frac{k}{n}\right)^{1 - d/D},$$

which is assumed by Theorem 10.

## A.7 Proof of Theorem 11

Let $P$ be a distribution on a manifold $M$ with density $f$. Let $\mathbf{X} = (X_1, \ldots, X_n)$ be the latent sample from $P$, and let $\mathbf{Y} = (Y_1, \ldots, Y_n)$ be the observed sample. The only fact that we use about the observed sample is that it is close to the corresponding latent sample point: $d(Y_i, X_i) \leq \theta$, where $\theta$ is the *noise radius*. We show that we can adapt the RSL algorithm to resolve $(\sigma, \epsilon)$ separated clusters $(A, A')$, provided that $\theta$ is sufficiently small compared to both $\sigma$ and $\epsilon$.

Again, we will pick values for $k, r, R$ based on a parameter $\rho$, defined as $\rho := \min(\frac{\sigma}{7}, \frac{\tau}{24}, \frac{\epsilon\tau}{144d})$.

**Lemma 20 (Separation)** *Suppose $k, r, R$ are chosen to satisfy*

$$\theta \leq r/2 \qquad r \leq \rho \qquad R := 5\rho,$$

$$v_d(r - 2\theta)^d(1 - \epsilon/6) \cdot \lambda \geq \frac{k}{n} + \frac{C_\delta}{n}\sqrt{k\mu},$$

$$v_d(r + 2\theta)^d(1 + \epsilon/6) \cdot \lambda(1 - \epsilon) \leq \frac{k}{n} - \frac{C_\delta}{n}\sqrt{k\mu},$$

*then, with probability $1 - \delta$, the following holds uniformly over all $(\sigma, \epsilon)$ separated clusters $(A, A')$:*

1. *If a latent sample point $X_i \in A_{M, \sigma - r + 2\theta} \cup A'_{M, \sigma - r + 2\theta}$, then the corresponding sample point $Y_i$ is kept during the cleaning step. If $X_i \in S_{M, \sigma - r - 2\theta}$, then $Y_i$ is removed.*

2. *The sets $\{Y_i : X_i \in A\}$ and $\{Y_i : X_i \in A'\}$ are disconnected in the graph $G_{r,R}$.*

**Proof.** To prove the first claim, suppose $X_i \in A_{\sigma - r + 2\theta} \cup A'_{\sigma - r + 2\theta}$. Consider the ball $B_M(X_i, r - 2\theta)$. It is completely inside $A_{M,\sigma} \cup A'_{M,\sigma}$, hence the density $f$ inside it is at least $\lambda$. Moreover, if $X_j$ is in $B_M(X_i, r - 2\theta)$, then by triangle inequality, we have

$$d(Y_j, Y_i) \leq d(X_j, Y_j) + d(X_j, X_i) + d(Y_i, X_i) \leq r.$$

Hence the ball $B(X_i, r)$ contains at least $k$ sample points, provided $B_M(X_i, r - 2\theta)$ contains at least $k$ points from $\mathbf{X}$. Finally, the true mass of the set $B_M(X_i, r - 2\theta)$ is at least

$$v_d(r - 2\theta)^d(1 - \epsilon/6) \cdot \lambda \geq \frac{k}{n} + \frac{C_\delta}{n}\sqrt{k\mu}.$$

Hence it contains at least $k$ latent sample points, and we are done.

Similarly, suppose $X_i \in S_{\sigma - r - 2\theta}$, and consider the ball $B_M(X_i, r + 2\theta)$. It is completely contained inside $S_{M,\sigma}$ and hence the density inside the ball is at most $\lambda(1 - \epsilon)$. Moreover, if $X_j$ is outside the set, then

$$d(Y_j, Y_i) \geq d(X_j, X_i) - d(X_i, Y_i) - d(X_j, Y_j) > r.$$

Hence the ball $B(Y_i, r)$ contains fewer than $k$ sample points, provided $B_M(X_i, r + 2\theta)$ contains fewer than $k$ points from $\mathbf{X}$. The true mass of the ball $B_M(X_i, r + 2\theta)$ is at most

$$v_d(r + 2\theta)^d(1 + \epsilon/6) \cdot \lambda(1 - \epsilon) \leq \frac{k}{n} - \frac{C_\delta}{n}\sqrt{k\mu}.$$

Hence the ball contains fewer than $k$ latent sample points, and we are done.

We now prove that the graph $G_{r,R}$ is disconnected. Suppose not. Then there must exist a sequence of latent sample points $x_0, x_1, \ldots, x_t \in \mathbf{Y}$ and a corresponding sequence of noisy sample points $y_0, \ldots, y_t \in \mathbf{X}$ such that $x_0 \in A$, $x_t \in A'$, and $d(y_{i-1}, y_i) \leq R$. Clearly $d(x_{i-1}, x_i) \leq R + 2\theta \leq \tau/4$. Thus the geodesic distance between $x_{i-1}$ and $x_i$ is less than $2(R + 2\theta)$. However, by the $(\sigma, \epsilon)$ separation condition, we must have a successive pair $(x_{i-1}, x_i)$ whose geodesic distance is at least $2(\sigma - r)$. This is a contradiction since we have set our parameters such that $2(\sigma - r) \geq 2(R + 2\theta)$. $\square$

**Lemma 21 (Connectedness)** *Assume that the conditions of Lemma 20 are satisfied. Then, with probability at least $1 - \delta$, the following holds uniformly over all $A$: if $\inf_{x \in A_{M,\sigma}} f(x) \geq \lambda$, then $\{Y_i \; : \; X_i \in A\}$ is connected in $G_{r,R}$.*

**Proof.** The proof is similar to that of Lemma 9, so we indicate only the necessary modifications, omitting the details. We now use a net of radius $(R - 2\theta)/4$, and the condition that $R \geq 4r$ is replaced by $R - 2\theta \geq 4r$. Finally, the $x_i$'s defined in the proof are latent sample points, whereas the algorithm observes an arbitrary point $y_i$ in a $\theta$-ball around the $x_i$. Thus, the distance between $y_{i-1}$ and $y_i$ is at most

$$4 \cdot \frac{R - 2\theta}{4} + d(y_i, x_i) + d(y_{i-1}, x_{i-1}) \leq R.$$

$\square$

In order to satisfy the conditions stated in Lemma 20, we need the assumption that $\theta$ is small compared to $r$. More precisely, we will assume that $\theta \leq r\epsilon/24d$. Under this assumption, we can satisfy the above conditions by ensuring that

$$v_d r^d (1 - \epsilon/12)(1 - \epsilon/6) \cdot \lambda \geq \frac{k}{n} + \frac{C_\delta}{n} \sqrt{k\mu},$$

$$v_d r^d (1 + \epsilon/6)(1 + \epsilon/6) \cdot \lambda(1 - \epsilon) \leq \frac{k}{n} - \frac{C_\delta}{n} \sqrt{k\mu}$$

As before, we can satisfy these equations by setting $k := O(C_\delta^2 \mu/\epsilon^2)$, and $r$ according to

$$v_d r^d (1 - \epsilon/12)(1 - \epsilon/6) \cdot \lambda = \frac{k}{n} + \frac{C_\delta}{n} \sqrt{k\mu}.$$

### A.8   Connection radius for polynomially bounded densities

In this section, we prove that in our algorithm (Figure 1), we can pick the connection radius $R$ to be $R := 4r$, independent of the other parameters, provided that the density level satisfies $\lambda \leq n^A$ for some absolute constant $A$. (Our original setting picked $R = 4\rho$ and $r \leq \rho$.)

More precisely, we will argue that the parameter $\mu$ in the algorithm can be safely replaced by a related parameter $\widetilde{\mu} := 2A \log n$ without affecting the performance of the algorithm. Pick $k = O(C_\delta^2 \widetilde{\mu}/\epsilon^2)$, and set $r, R$ by the equations

$$v_d r^d \lambda = \frac{1}{1 - \epsilon/6} \left( \frac{k}{n} + \frac{C_2 \log(1/\delta)}{n} \sqrt{k\widetilde{\mu}} \right),$$

$$R = 4r.$$

The crucial ingredient in the analysis of our algorithm is the uniform convergence property of balls centered at the sample points and net points (Lemma 6), so we first verify that this statement remains true. Note that by our choice of $r$, we have

$$v_d r^d \lambda \geq \frac{k}{n} \geq \frac{1}{n},$$

so that $1/r^d \leq v_d n\lambda \leq v_d n^{A+1} \leq n^{A+1}$ (since $v_d < 1$ for sufficiently large $d$). As before, we consider a net $\mathcal{N}$ of radius $R/4$ (i.e., $r$); by Lemma 16, size of this net is at most $c^d/r^d$ for some absolute constant $c > 0$. Thus by Lemma 6, we have the uniform convergence property, provided the parameter $\mu$ is replaced by

$$\log n + \log |\mathcal{N}| = \log n + \log(1/r^d) + O(1) = (A + 2) \log n + O(1).$$

Notice that $\widetilde{\mu}$ is picked to be a safe upper bound on this quantity, hence the lemma holds when $\mu$ is replaced by $\widetilde{\mu}$.

Finally, it is easy to check that our choice of parameters satisfies all the conditions given in the separation lemma. Hence the separation and connectedness guarantees (Lemmas 8 and 9), together with their proofs, remain unaffected.