[Reviews · NeurIPS 2013]

Submitted by Assigned_Reviewer_4

The paper discusses the estimation of the cluster tree when the probability density function is supported on a d dimension manifold in a D dimensional space. They show that the algorithm RSL proposed in (1) is consistent and the convergence rate, very roughly, depends on d-the dimension of the manifold and not on D - the dimension of the ambient space (but then the convergence rate also depends on tau-the conditional number and an epsilon^{d+2} factor instead of an epsilon^2 factor).

The main result is achieved by repeating the technique in (1). To do that, the authors had to show: First,a bound on a size of an s-net in the manifold setting. Second, bounds on the deformation of the volume (i.e. that B(x,r)cap M has roughly the volume of a d-dimensional ball of radius r where d is the dimension of M). The authors are able to show both under the assumption of small conditional number.

I think it is interesting to know how convergence rate changes under this assumption (i.e. manifold assumption) and the paper give both lower bounds and upper bounds that are not trivial. So even though the convergence rate depends on sizes that are not available (the dimension of the manifold and the conditional number), still the results are interesting.

I found the writing very unclear and certain definitions are even confusing:
*******************************
1) The statement in thm.4 is wrong. A much stronger statement is proved in thm.6 of [1] than def.3-consistency. (see also, remark after thm.6 in [1]).

Theorem 6 states that with high probability: uniformly, every A A' that satisfy (\sigma,\epsilon)-separation: we get separation and conectedness.

Theorem 4 states that for every A A' that satisfy (\sigma, \epsilon)-seperation, with high probability we get separation and connectedness.

These are not equivalent statements. Please correct this.
*****************************
2) Still in definition 3: what is n? who is C_n (is it a random variable? how is it defined? is it different than hat{C}) I had to rely on the definition in (1) to understand what is meant here.
3) When using Lemma 16: It is worth indexing and mentioning which inequality is used and how at each step. Not all steps are clear, it seems that at last step you use
(1+4r/t(1+4r/t))(1+4r/t) < (1+6r/t) but that's not even hinted. The steps should be clearer.

4) Lemma 18:
I think a 1/2 is missing from the definition of v_cap.
Worth mentioning that Gamma(1/2)=sqrt(pi) otherwise it's not clear where it went.


Further suggestions:

The lower bound you produce depends on the conditional number, it might be worth mentioning the lower bound you produce are not an improvement over the lower bound in (1), but are different (e.g. in a linear subspace that has 1/tau=0 your lower bound is meaningless while (1) gives a sensible lower bound).

Regarding the parameter \rho, does it really make sense to choose salience parameter 2\sigma > tau? won't it be easier to simply assume (3\sigma/16) < (\tau/16)?
Summary: The authors demonstrate how one can generalize results to the manifold case by having interesting bounds on s-net. I found the paper not clear enough, and definition 3 is wrong as far as I can see.

Submitted by Assigned_Reviewer_5

This paper analyzes the robust single linkage algorithm for finding the cluster tree when the support of the density lies on a manifold. Previous work [1] proposed and analyzed the algorithm for the same, when the density is supported on the entire space. This paper shows that the same algorithm could be used in the manifold case with rates depending just on the manifold dimension rather than the ambient dimension. The proof flow is very similar to that of [1] with several modifications to handle the fact that the density is actually supported on a manifold.

The main point conveyed by the paper seems to be not emphasized sufficiently - despite the fact that the density lies on a manifold, the same RSL algorithm that is based on euclidean distance achieves rates that depend only the manifold dimension - giving some intuition for this fact (say even with a simple synthetic example) might help the reader a lot. Since the proofs are more or less based on similar arguments as that of [1], it is not clear what is the fundamental idea that is behind this phenomenon- at least in the way the proofs are presented.

I went over the proofs and it seems ok to me. More discussion about the parameter $\rho$ might be helpful to the reader - I guess the previous point is related to this fact. Also I do not really agree with what authors call the 'class of RSL' algorithm (which has some consequences in terms of understanding the implications of the lower bound). Specifically what does ' of the form described in the algorithm in Figure 1' mean - isn't just this one algorithm in that case ?

Summary: This paper analyzes the robust single linkage algorithm for finding the cluster tree when the support of the density lies on a manifold and shows that RSL algorithm that is based on euclidean distance achieves rates that depend only the manifold dimension.

Submitted by Assigned_Reviewer_7

This paper presents an analysis of a k-nearest neighbor based algorithm recently proposed by Chaudhuri and Dasgupta for the case where the data is on or concentrated on a manifold. The key result is that the convergence rate depends on the dimension of the underlying manifold. To obtain the result on the manifold the authors adapt the theory used in Chaudhuri and Dasgupta to the manifold case using the sampling tools developed in Niyogi, Smale, Weinberger. In the case where the data is concentrated on a manifold the use the tools developed by a serious of papers by Rinaldi and others.

The theory seems correct and these are nice results. The algorithm analyzed may be more appropriate for stratified spaces than manifolds but this is a more minor point. The one negative about this paper is that the algorithm is not novel and many of the theoretical tools used are not novel. However, this complaint can be made of many theoretical papers in NIPs.
Summary: The authors present a theoretical analysis of cluster trees on manifolds and show the rate is a function of the manifold.
Author Feedback

Author rebuttal: We would like to thank the reviewers for their comments and valuable feedback. We will revise the manuscript to address some of the reviewers concerns and we briefly discuss these concerns here.

Reviewer 5 makes some excellent suggestions and we will incorporate a longer discussion of these in the revised version of the manuscript. The main intuition for why the RSL algorithm is able to take advantage of the manifold structure and achieve rates that depend only on the manifold dimension is that when the density is supported on or near a lower dimensional manifold, balls around sample contain many more points than one would expect if the manifold hypothesis were not true. This is for instance reflected in the k-NN distance which for a point in a region of density \lambda intuitively behaves as (k/(nv_D \lambda))^{1/D} for full dimensional densities but is much smaller, i.e. (k/(nv_d \lambda))^{1/d}, for densities on or near a well conditioned d-dimensional manifold. This lets one resolve the cluster tree from fewer samples. We will describe this in more detail and include a synthetic example in the paper to help build intuition.

The rates of convergence for the RSL algorithm are fundamentally determined by the radius of the largest ball that one can use in resolving the clusters. In the Euclidean case the separation of the clusters (\sigma) constrains the largest ball. In the manifold case this radius is determined by three distinct effects. The parameter \rho is determined (ignoring constants and d terms) by the minimum of \sigma, \tau, and \epsilon \tau/d (notice that since d can be 0 the last term is not dominated by the first). \sigma is as before. \tau is a spatial effect because of the curvature of the manifold and \epsilon \tau/d is a density effect. Intuitively using larger balls than \tau can cause a spatial chaining across separate regions of the manifold, and using balls of radius larger than \epsilon \tau/d means the RSL algorithm cannot distinguish densities of \lambda and \lambda(1-\epsilon) in regions of different curvature.

Regarding the terminology, by "the class of RSL algorithms", we really meant the RSL algorithm with any choice of the various tuning parameters. It is however easy to see that the lower bound for instance also applies to algorithms that output the cluster tree of a kernel density estimate. We will clarify this further.

We thank Reviewer 4 for a careful reading of the manuscript. The reviewer had several complaints regarding the writing which we now address.
(1) Regarding the uniformity over sets A and A', we would like to first emphasize that (modulo the typo from the second point below) the definition is identical to the one used in the paper of [1] and the results in our paper are not weaker than those in [1]. The results are not uniform over clusters A and A' in the sense that for a given value of n, there can be clusters A and A' that are not resolved (ones that have \sigma and \epsilon small enough as a function of n, this is clear since the rates depend on \sigma and \epsilon). They are however uniform over the class of \sigma, \epsilon clusters, i.e. for a fixed value of \sigma and \epsilon there is an n large enough so that all \sigma, \epsilon clusters are resolved (no matter how many such
pairs there are). This is identical to the situation in [1] and the results we have are not weaker in this sense. We will clarify this point further.
(2) We will change C_n to \hat{C}.
(3) We will clarify the steps.
(4) We have checked and there is no 1/2 missing, the reviewer is possibly confusing the formula for volume with the one for area (we are calculating the d-dimensional volume of a spherical cap which in standard terminology is the area of the spherical cap on S^d). We will clarify this.

Regarding the two additional suggestions. We will clarify the relation between our lower bound and the lower bound in [1], as the reviewer pointed out they are not directly comparable. Our lower bound is intended to be complementary to the one in [1] in that it explains a different aspect of the \rho parameter.

Regarding whether it makes sense to choose 2\sigma > \tau, we should emphasize that \sigma is in some sense a user specified parameter and \tau is an intrinsic one. \sigma can be much larger than \tau for instance if the manifold consists of two poorly conditioned pieces that are well separated, and the user wishes to only resolve the pieces.

We thank Reviewer 7 for the valuable feedback. Reviewer 7's main concern is regarding the novelty of the techniques used in the paper. While it is true that we base our main proof and theorem on the work of [1] (and others), we should point out that the results of [1] have only trivial implications for the cases we consider when the density is singular. We would also like to emphasize a few novel aspects of our work.
(1) We use different arguments to show both connection and separation, the choice of tuning parameters is different because we need to account for the curvature of the manifold in various places, and the core large deviation inequality while similar in form is proved using fundamentally different techniques.
(2) We would also like to emphasize that our lower bound is completely different in both its proof and implication.
(3) We have in this paper initiated the study of noisy cluster tree recovery. To study the noisy case we have introduced two new natural notions of consistency -- in the clutter noise case we consider consistency of only the samples generated from the "true" distribution and in the case of additive noise we consider consider consistency of the latent (before contamination by additive noise) samples. In each case we specify conditions under which we can resolve the whole or parts of the cluster tree.